# Expanding two-way texting for post-operative follow-up: A cost analysis of the implementation and scale-up in routine voluntary medical male circumcision settings in South Africa

**Molly Unsworth**[1,2], **Isabella Fabens**[1,2], **Geoffrey Setswe**[3,4], **Khumbulani Moyo**[5], **Jacqueline Pienaar**[3], **Calsile Makhele**[3], **Motshana Phohole**[5], **Nelson Igaba**[5], **Sizwe Hlongwane**[5], **Maria Sardini**[3], **Tracy Dong**[6], **Monisha Sharma**[2], **Hannock Tweya**[1,2], **Felex Ndebele**[3], **Marianne Holec**[1,2], **Caryl Feldacker**[1,2]*

1 International Training and Education Center for Health (I-TECH), University of Washington, Seattle, Washington, United States of America, 2 Department of Global Health, University of Washington, Seattle, Washington, United States of America, 3 The Aurum Institute, Johannesburg, South Africa, 4 Department of Health Studies, University of South Africa, Pretoria, South Africa, 5 Right to Care, Pretoria, South Africa, 6 Fred Hutchinson Cancer Center, Seattle, Washington, United States of America

* cfeld@uw.edu

## Abstract

Up to 98% of adult voluntary medical male circumcision (VMMC) clients heal without adverse events (AEs) in South Africa and in the sub-Saharan Africa region. Yet, all clients in South Africa are required to attend in-person reviews, creating added effort for providers and clients. A randomized controlled trial (RCT) using our fee-free, open-source, two-way texting (2wT) approach showed that males could independently monitor their healing with nurse-led telehealth support. 2wT was more cost-effective than routine visits for quality post-operative monitoring. The objectives of this study were:1) assess the additive cost of 2wT vs. standard of care (SoC) during a stepped wedge design (SWD) expansion trial; 2) determine the cost of augmenting 2wT implementation with dedicated personnel during peak VMMC periods; and 3) estimate the cost savings of 2wT from the payer perspective if scaled in routine settings. Data were collected from routine financial reports and complemented by previous RCT time-motion estimates. We conducted activity-based costing of SWD and peak season periods. Sensitivity analysis to estimates 2wT costs at scale. Data included 6,842 males; 2,586 (38%) opted for 2wT. 2wT participants attended an average of zero in-person visits; SoC males had an average of 2 in-person visits. Under 2wT, quality care improved: AE ascertainment increased while loss to follow-up decreased. Given a VMMC population of 10,000 adults, scenario analysis suggests that: 1) 2wT becomes cost neutral with 45% 2wT enrollment; 2) 2wT saves $0.29/client with 60% 2wT enrollment; and 3) 2wT saves $0.46/client with 80% 2wT enrollment. When scaled, 2wT appears to significantly reduce healthcare system costs while improving the quality of post-operative care without additional client costs. Further scale-up of 2wT for eligible males

**Data availability statement:** All data for the study is provided in supplementary information (Table 1)

**Funding:** Research reported in this publication was supported by the National Institute of Nursing Research (NINR) of the National Institutes of Health under award number 5R01NR019229, "Expanding and Scaling Two-way Texting to Reduce Unnecessary Follow-Up and Improve Adverse Event Identification Among Voluntary Medical Male Circumcision Clients in the Republic of South Africa." Statistical support was provided by the University of Washington and its Fred Hutch Center for AIDS Research, a National Institutes of Health (NIH)-funded program under award number AI027757 that is supported by the following NIH Institutes and Centers: National Institute of Allergy and Infectious Diseases, National Cancer Institute, National Institute of Mental Health, National Institute on Drug Abuse, Eunice Kennedy Shriver National Institute of Child Health and Human Development, National Heart, Lung and Blood Institute, National Investigation Agency, National Institute of General Medical Sciences, and National Institute of Diabetes and Digestive and Kidney Diseases. The content is solely the responsibility of the authors and does not necessarily represent the official views of the National Institutes of Health.

**Competing interests:** The authors have declared that no competing interests exist.

across VMMC and other post-operative contexts in South Africa would likely increase cost savings while dramatically reducing the burden of in-person visits on patients and clinics.

## Introduction

For more than two decades, studies on adult voluntary medical male circumcision (VMMC) have demonstrated that VMMC is an efficient, safe, and cost-effective method to reduce the continued burden of HIV [1–3]. Rates of moderate and severe adverse events (AEs) are less than 2% in VMMC programs at scale but multiple in-person post-operative visits within 14 days of the procedure continue to be recommended for all clients. In South Africa, despite few AEs, national guidelines require two in-person follow-up visits after VMMC, a largely unnecessary burden for most patients and their providers that may deter VMMC uptake [4,5].

Mobile health (mHealth) interventions in this context therefore offer an innovative way of utilizing digital technologies to reduce the burden of in-person visits for clients and healthcare workers. mHealth interventions can help the digitization of overburdened healthcare systems, as well as assist with reaching remote or underserved areas by providing remote access to health services [6]. Two prior randomized clinical trials (RCTs) were conducted in routine VMMC clinics in Zimbabwe and South Africa, piloting a novel two-way texting (2wT) approach between eligible males ages 15-years and older and VMMC nurses [5,7]. Both studies demonstrated that 2wT is safe, reduced follow-up workload, and is cost-effective compared to routine VMMC post-operative visits. However as both studies were done within the context of an RCT, the results did not explore the true costs of 2wT when implemented as part of routine VMMC care.

To further evaluate the effects of 2wT in routine VMMC programs operating in South Africa, the International Training and Education Center for Health (I-TECH) at the University of Washington, The Aurum Institute, Right to Care, and Medic built upon their RCT evidence by conducting a quasi-experimental, stepped wedge design (SWD) expansion study across three districts: an urban district in Gauteng province and both a peri-urban and a rural district in the North West province. The expansion study was conducted in two phases, intensive and maintenance, and 2wT was implemented as part of routine VMMC service delivery. Eligible males could opt-into 2wT follow-up as an alternative to scheduled, post-operative visits. During the intensive phase (January – October 2023 with follow-up through December 2023), sites continued to receive support from the research team, and the 2wT system continued to undergo revisions to increase usability in the routine setting. During the maintenance phase (January 2024, onward), sites received quality assurance support from the study team, preparing them to fully maintain the 2wT approach independently by the end of 2024. Results on clinical outcomes from the SWD that demonstrated 2wT safety and effectiveness were published separately [8].

Despite cost often being identified as a key outcome, there remains a lack of transparency and comprehensive cost evaluation of mHealth interventions [9,10]. Scale-up of previous mHealth interventions in low- and middle-income countries (LMIC) has failed due to a lack of data on the cost of programs at scale [11]. While the evidence base has grown in recent years, there remains a need for high-quality economic evaluation to better understand the complexities and implications of scaling up digital health strategies [12]. Further evidence of economic evaluation using an implementation science lens is needed to support the scale-up of mHealth interventions in routine practice.

Therefore, to complement the safety and effectiveness evidence from the SWD, we embedded a costing study with three objectives: (1) assess the additive cost of implementing 2wT for

follow-up during the intensive phase of the stepped wedge design SWD, (2) measure the costs of augmenting 2wT personnel to reach high 2wT demand during peak VMMC season, and (3) estimate the cost savings of 2wT compared to routine VMMC follow-up at scale from the payer perspective. By looking at costs of 2wT when implemented in routine VMMC clinics without RCT conditions, we aim to provide updated estimates on the costs associated with the use of 2wT in routine practice, during common VMMC campaign approaches, and inform the feasibility of 2wT expansion in routine practice.

## Methods

### Background

**2wT intervention.** The open-source 2wT technology and implementation approach for VMMC follow-up has been previously described in detail [7,13,14]. Briefly, the 2wT approach provides an option for clients to choose interactive, text-based telehealth with a dedicated 2wT VMMC nurse in lieu of in-person visits the standard of care (SoC), for post-VMMC follow-up. The hybrid 2wT system combines automated education messages along with messages requesting a client response to confirm appropriate healing. 2wT allows for clients to interact with a clinician who can provide reassurance, wound care instructions, and refer them to in-person review for potential complications during the critical 14-days post VMMC procedure. 2wT participants can always seek in-person care when desired. 2wT does not require clients to have a smartphone; clients can use any basic mobile phone with SMS capability and communicate in various local languages.

For clinics, 2wT participants are managed via the open-source app that is downloaded onto smartphones, tablets, or desktop computers. Enrollment of clients in 2wT, education on using the 2wT system, and 2wT-specific post-operative counseling, as well as any needed or requested in-person follow-up visits for 2wT participants all take place at routine VMMC clinics ("spokes"), with care provided by existing VMMC teams. The 2wT dedicated nurse ("hub"), is the first line of communication with 2wT participants from all sites, providing clinical care via messaging, conducting initial phone tracing for clients who failed to respond to texts, and referring clients back to their 2wT enrollment VMMC clinic "spoke" for follow-up care if needed. The 2wT system includes quality assurance prompts. For the hub, if a client does not respond by day 3 after VMMC, the hub nurse receives a tracing "Task" to follow up that client by phone, clearing the Task by documenting whether the client was reached and ok; reached and referred to care; or not reached so referred to the site for tracing (unknown client status). For the sites (spokes), clients referred to care from the hub generate a "Task" for them to complete after client review, noting whether the client was 1) okay (no AE); 2) was confirmed with an AE; or 3) client did not return (noting client stable/not stable/unknown).

**Stepped wedge design (SWD): Implementation and results.** The expansion study was implemented across study sites in two phases using the hub-to-spoke model. During the intensive SWD phase, routine VMMC clinics acted as the spokes and received support from the 2wT research team who provided training, quality assurance, and clinical oversight while also maintaining the centralized 2wT hub. Between January to December 2023, in the SWD, 2,856 of 6,842 clients opted into 2wT (37.8%) across three intervention waves and two platforms (SMS or WhatsApp). Among those with post-operative follow-up, the AE ascertainment rate was higher among 2wT participants (0.60%) than SoC clients (0.13%) (p = 0.0018). On average, 2wT participants had 2.1 fewer visits compared to SoC clients (p<0.001). Among 2wT participants, 1832/2586 (71%) responded via 2wT by Day 3 and 2,069/2,586 (80%) responded via 2wT over 14 days. Lost to follow-up (LTFU), or no response or visit, was documented for 93 2wT participants (3.6%) and 342 (8.0%) SoC clients [8].

Following the completion of the intensive phase, the study transitioned to a maintenance phase, starting January 1, 2024. Thereafter, the study-supported 2wT team was gradually withdrawn, with the aim of fully transitioning the 2wT system and management to local teams over an additional year of implementation.

**2wT enrollment threshold: Identifying a 2wT quality gap during VMMC campaigns.** In South Africa, like other high priority VMMC countries in sub-Saharan Africa (SSA), VMMC clinics experience seasonal fluctuation: the months of June and July are often high-volume VMMC campaign periods, with smaller campaigns in March and September. A large proportion of annual circumcision productivity takes place in these higher demand periods, often with focus on reaching adolescent boys ages 15 and older who are on school holidays [15,16]. During the intensive SWD phase, the 2023 VMMC campaign period saw 2wT enrollments spike from a weekly high of approximately 200 to over 450, resulting in approximately 900 clients simultaneously in the 2wT system in the first 14-days. This high-volume VMMC period and surge in 2wT enrollment created an extraordinary workload for the 2wT hub, resulting in delays in completing the tracing Task for 2wT participants who had not responded to 2wT texts by day 3. **Fig 1** illustrates that during the 4-week campaign period,

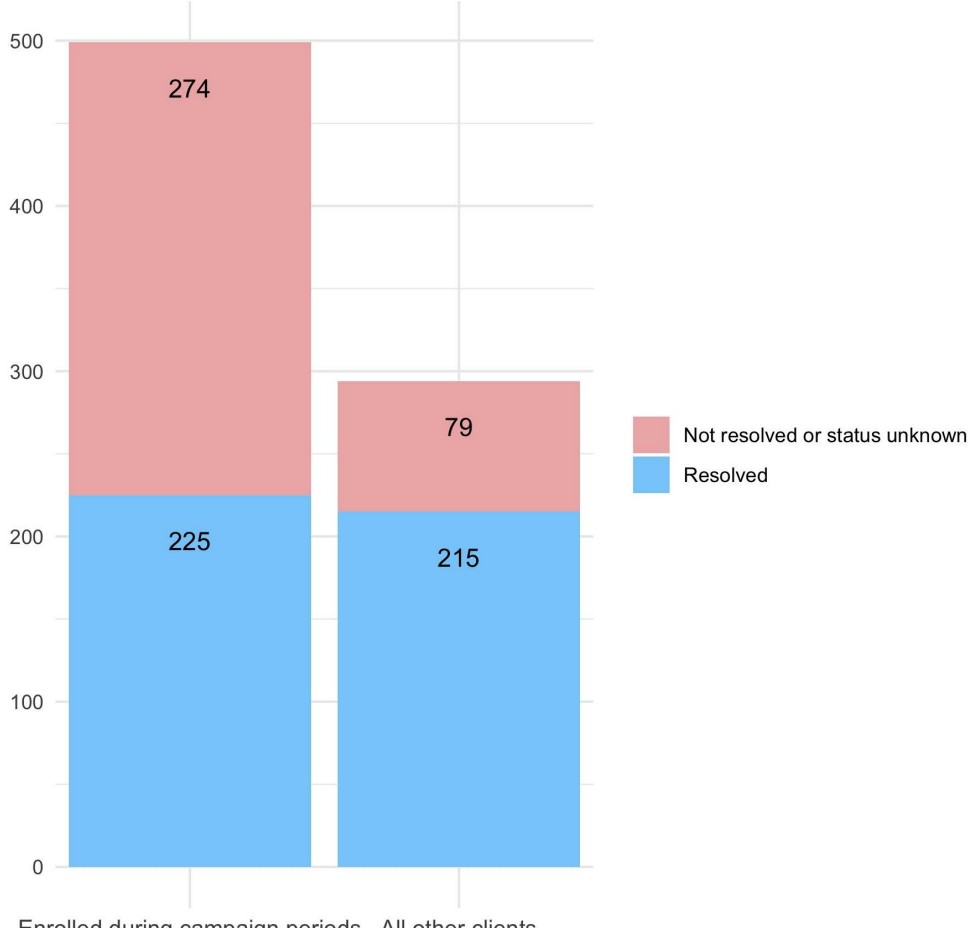

**Resolution of potential LTFU tracing task**

**Fig 1. Resolution of tracing tasks by the hub for 2wT participants during campaign periods vs. routine VMMC levels, South Africa, 2023.**

including only clients enrolled from mid-June through mid-July 2023, the total number of 2wT participants who did not respond by day 3 after VMMC was 499 of 1,464 (34%) of which 274/1,464 (19%) were not successfully traced by the hub. During the non-campaign months, 294/1,122 (26%) of 2wT participants did not respond by day 3, but only 7% of the Tasks did not result in successfully tracing by the hub. This comparison suggests that there is a 2wT participant to hub threshold at which the quality of 2wT-based care may diminish. A forthcoming paper will examine the 2wT tracing and referral cascade, including this gap in 2wT quality when the threshold is reached.

## Costing methodology

We performed a micro-costing analysis of 2wT in routine settings from the perspective of the program implementer. Costs to VMMC clients were excluded. The methods used followed the guidance of The Consolidated Health Economic Evaluation Reporting Standards (CHEERS) statement as described in the supplementary materials (S2 Table) [17].

Primary data collected during the intensive and maintenance phases of the SWD were used to estimate the additive cost of 2wT and the costs of increasing staff support during campaign periods. To estimate the cost savings compared to in-person VMMC follow-up, we used data on routine care that were collected during the earlier RCT as data on routine clinic-based follow-up were not collected during the SWD study period. The costing analysis was conducted in Microsoft Excel. An editable costing tool (S1 Table) captures all data and costing algorithms. All parameters are adjustable, allowing for future costing studies to evaluate the cost of 2wT scale-up in other settings. Costs were collected in South African rand (ZAR) and United States dollar (USD). Results are financial costs reported in 2023 U.S. dollar values (USD).

## Objective 1: Additive costs of implementing 2wT in routine VMMC care

Cost data were taken from program financial records, RCT study logs, SWD study logs, and inputs from study staff. Cost parameters from the study period are presented in **Table 1**. Costs collected in South African Rand (ZAR) were converted to United States Dollar (USD) using the average exchange rate over the study period of ZAR 18.577 = $1.00 USD. Costs were grouped into four distinct categories (**Table 1**): personnel costs, recurrent costs, variable costs, and capital costs. Personnel costs included the salaries of the hub nurse, project administrator, and information technology (IT) professional for 2wT system up-keep. Recurrent costs included monthly fees associated with the maintenance and usage of the 2wT system, including reverse billing to cover all client messaging costs, as well as the cost of airtime for the hub nurse. Variable costs included the cost of client SMS messages, which was calculated by

**Table 1. Cost categories and descriptions of costs from the intensive phase of SWD that were included in the cost analysis.**

| Cost Category | Description |
|---|---|
| Recurrent costs | Monthly costs of fees associated with hosting and maintaining the 2wT system (server hosting 2wT system, SMS gateway integration, API fee) and airtime for hub nurse. |
| Variable costs | The cost of all SMS messages sent by 2wT participants and the hub nurse. |
| Capital costs | One-time cost of laptop and mobile phone for hub nurse. |
| Personnel costs | The staff salary of a project administrator, 2wT-specific hub nurse, and IT professional for 2wT system maintenance. |

Abbreviations: SMS, short-messaging system; API, application programming interface; 2wT, two-way texting; IT, information technology.

multiplying the unit cost of a single text message by the number of texts sent per client during the study period. Capital costs were the equipment needed for 2wT client management by the hub nurse, including a laptop and mobile phone. Capital costs were annualized assuming a 5-year useful life and discounted annually at 3%, the standard recommended for economic evaluation in LMICs [18]. The total cost of the intervention over the study period was divided by the number of 2wT participants to determine the cost per client.

## Objective 2: Costs of augmenting 2wT during campaign periods

To evaluate the additive personnel cost needed to ensure quality 2wT implementation during the busy season campaign, we used cost data from the 2024 pilot program. We calculated the total cost and cost per client of meeting campaign period demand for 2wT. Personnel costs included the monthly stipend of the intern and the increase in full-time equivalent (FTE) for the hub nurse. Additional costs were the tablets and airtime used by each intern. Tablet costs were annualized assuming a 5-year useful life and discounted annually at 3%. Based on data collected during the 2023 and 2024 campaign periods, we assumed a total rate of 500 new clients per week during the busy season across all sites. We calculated the monthly cost per site of the pilot program by summing the additional personnel costs and annualized cost of one tablet for 5-weeks. The total cost of implementing the pilot program for a 5-week program was calculated by adding the additional cost of increasing the hub nurse FTE along with the costs associated with hiring one intern per site. The total cost per client was calculated by dividing the total cost per category by the total number of expected clients for the 5-week period.

## Objective 3: Estimating cost savings of 2wT implementation in routine clinical practice

Costing estimates from the intensive phase and subsequent pilot program for the 2024 busy season were used along with prior data collected during the RCT to estimate the per client cost savings of implementing 2wT in routine VMMC clinics. Data collected during the intensive phase were used for parameter estimates on the average number of follow-up visits, reported rate of moderate/severe AEs, LTFU rate of clients, and salaries of 2wT personnel. Parameter estimates collected during the RCT were used to determine the cost of follow-up visits and tracing. As data were not available during the SWD on the number of routine and 2wT participants for which tracing was initiated, the cost of client tracing assumed that all clients who were LTFU had three phone calls and three home tracing attempts, as is standard.

### Sensitivity analysis

One-way univariate sensitivity analyses were used to assess the most influential parameters for cost savings of 2wT compared to routine VMMC follow-up and evaluate their impact. In the base-case scenario, enrollment rate in 2wT was based on the enrollment levels during the intensive phase of the SWD. All parameters were varied using a range of +/- 20%.

### Ethics

The protocol for the scale-up of 2wT as part of the SWD was approved by the review boards of the University of Washington (Study 00009703; PI Feldacker) and the University of Witwatersrand, Human Research Ethics Committee (No. 200204; PI Setswe). The funders had no role in study design, data collection, analysis, decision to publish, or preparation of the manuscript. No informed consent was undertaken for this follow-up approach as 2wT was implemented as part of routine VMMC procedures for males ages 15 and older and no personal identifiers were used in these analyses.

## Results

During the intensive intervention phase of the SWD, 6,842 males aged 15 years or older underwent VMMC: 2586 (38%) opted to participate in 2wT for follow-up. From the SWD clinical outcomes, we found that on average, 2wT participants had 0.07 visits (SD 0.30) compared to 2.04 visits among routine clients (SD 0.75) [8]. Routine clients had a higher rate of loss to follow-up (LTFU) at 8% compared to 2wT participants at 4% [8].

### Objective 1: Additive costs of implementing 2wT in routine VMMC care

During the intensive phase of SWD, 2wT-specific activities provided an additional cost per client of $9.53: $5.39 was due to research-specific costs and $4.14 was attributable to routine 2wT administration (**Table 2**). Of the $4.14 increased cost of routine administration: $0.50 were recurrent costs related to the 2wT technology, $0.26 were variable costs of SMS messaging, and $3.38 were personnel costs.

### Objective 2: Costs of augmenting 2wT during campaign periods

The pilot program to augment 2wT personnel to ensure quality 2wT during campaign periods cost $1,157.37 per month for one site, with $857.15 to increase the FTE of the hub nurse and $284.96 for an additional 2wT-specific intern (**Table 3**). We use the average of 500 clients/week, combined, across 3 sites observed in the 2023 and 2024 campaign periods. Implementing the 2wT staff augmentation for a one-month campaign period at this enrollment level would increase the cost per client by $0.81.

### Objective 3: Estimating cost savings of 2wT implementation in standard (non-research) clinical practice

The parameters used to calculate the cost savings of 2wT compared to routine VMMC follow-up are included in **Table 4**. For a district performing 10,000 VMMCs annually, enrolling clients in 2wT at the rate observed during the intensive phase of the SWD (40%) would increase costs by $0.06 per client. 2wT becomes cost neutral with 45% enrollment into the intervention. Increasing enrollment to 80% would reduce the mean cost by $0.46/client. Further scaled up to a scenario where 100,000 VMMCs are performed annually, at 40% enrollment in 2wT costs of follow-up would be reduced by $0.88/client, decreasing by $0.93/client at 80% enrollment.

The base-case compared the cost per client of 2wT to routine in-person follow-up, assuming implementation at the level observed during the intensive phase of the SWD. We assumed 10,000 annual VMMCs and an enrollment rate of 40% in 2wT, resulting in an increased cost of $0.06/client in the base-case scenario. In the univariate sensitivity analysis (**Fig 2**), the cost savings of 2wT for post-VMMC follow-up compared to routine in-person visits was most sensitive to the enrollment rate in 2wT, average number of follow-up visits attended by routine clients, and minutes spent by clinicians at post-operative visits.

## Discussion

When implemented in routine VMMC clinical practice at small scale, 2wT follow-up increased AE ascertainment and reduced LTFU, increasing the quality of post-operative care. At small scale, 2wT also raised costs by approximately $4/client – a modest additional cost to the payer balanced by fewer in-person visits, increased AE ascertainment, and decreased rates of LTFU. However, when operating at scale in a district performing a reasonable 10,000 VMMCs annually, scenario analysis suggests that 2wT becomes cost neutral with 45% 2wT

**Table 2. (A) Cost parameters and (B) cost per client of 2wT for VMMC follow-up during the intensive stepped wedge implementation, January 30 – November 2, 2023.**

**A.**

| Routine-administration for 2wT | | | |
|---|---|---|---|
| **Cost Category** | **Input** | **Range (if applicable)** | **Source** |
| **Recurrent Costs** | | | |
| Africa Is Talking (API) | $107.66 | | Study budget report |
| SMS bundles (TextIT) | $25.00 | | Study budget report |
| Hub nurse airtime/data | $10.77 | | Study budget report |
| *Variable Costs* | | | |
| Cost per SMS | $0.01 | | Study budget report |
| Number of outgoing texts | 13 | | Study log |
| Number of incoming texts | 7 | | Study log |
| *Personnel Costs* | | | |
| Project administrator wage (per month) | $520.37 | | Aurum |
| *FTE for 2wT* | *20%* | | *Study log* |
| Hub nurse wage (per month) | $1,426.47 | $1,345.73 - $1,507.22 | Aurum |
| *FTE for 2wT* | *40%* | | *Study log* |
| IT professional wage (per month) | $4,312.50 | $4125 - $4500 | I-TECH |
| *FTE for 2wT* | *6.7%* | | *Study log* |
| **Research-specific** | | | |
| **Cost Category** | **Input** | **Range (if applicable)** | **Source** |
| *Recurrent Costs* | | | |
| Server hosting (Amazon Web Services) | $300.00 | | Study budget report |
| *Capital Costs* | | | |
| Laptop | $349.89 | | Study budget report |
| Cellphone | $16.15 | | Study budget report |
| *Personnel Costs* | | | |
| Project administrator wage (per month) | $520.37 | | Aurum |
| *FTE for 2wT* | *80%* | | *Study log* |
| Hub nurse wage (per month) | $1,426.47 | $1,345.73 - $1,507.22 | Aurum |
| *FTE for 2wT* | *60%* | | *Study log* |

**B.**

| Cost Category | Cost per Client |
|---|---|
| **Routine-administration** | |
| Recurrent costs | $0.50 |
| Variable costs | $0.26 |
| Personnel costs | $3.38 |
| **Total** | **$4.14** |
| **Research-specific** | |
| Recurrent costs | $0.26 |
| Capital costs | $0.02 |
| Personnel costs | $5.10 |
| **Total** | **$5.39** |
| **Total cost per client (n = 2586)** | **$9.53** |

Table 3. Cost estimates for 2wT support staff scale-up for a 1-month VMMC campaign period.

| Cost Category | Monthly cost* (per site) | Cost per client (n = 2,000) |
|---|---|---|
| **Personnel costs** | | |
| Centralized EN nurse 100% | $857.15* | |
| Additional project intern 100% | $284.96 | |
| | **$1,142.12** | **$0.79** |
| **Recurrent costs** | | |
| Airtime for project intern | **$10.78** | **$0.01** |
| **Capital costs** | | |
| Annualized equipment | **$4.24** | **$0.01** |
| **Total** | **$1,157.37** | **$0.81** |

Abbreviations: SMS, short-messaging system

*Hub nurse salary would be shared across multiple sites in hub-spoke model

enrollment; saves $0.29/client with 60% 2wT enrollment; and saves $0.46/client with 80% 2wT enrollment without consideration of the cost impact of improved quality, monitoring, and client engagement in care. During 4-week campaign periods, to ensure 2wT quality and adherence to protocols, adding temporary, dedicated 2wT personnel would cost an additional $0.81/client. Across all 12 months, even with additional human resources during peak VMMC operations, it again appears that 2wT becomes cost neutral when implemented on larger scales or with increased enrollment rates. This strongly suggests that 2wT could lead to significant health system savings. These results align with prior costing studies on 2wT for VMMC follow-up in Zimbabwe and South Africa which found 2wT to be a cost-effective method to increase AE ascertainment and improve the quality of post-operative care at scale.

While our results differ from what was previously reported in the 2022 2wT RCT in South Africa [19], they provide an updated cost estimate for routine 2wT implementation. The RCT costing analysis found that 2wT could save $3.56 per client. In this SWD, we suggest that enrolling 4,000 2wT participants per year in a district that performs 10,000 VMMCs annually would only add a minimal $0.06 per 2wT client, with increased cost savings when a greater proportion of clients enroll. The difference in cost savings between RCT and more routine SWD conditions is likely due to several key implementation factors. First, the RCT did not include additional costs of maintaining the 2wT system, recurrent monthly technology fees, or equipment needed for systems operation, which were included in this analysis. Second, the RCT estimated costs separately in rural and urban areas, and found more cost savings in rural areas, while in the SWD, the clients were predominantly urban. Lastly, the RCT included time-in-motion observations to allocate time, and therefore cost, more accurately (e.g., client tracing activities, additional clinic costs for visits (reception, clerk), an activity that we could not undertake in this small, embedded costing study.

Personnel costs make up the largest proportion of 2wT-specific costs, demonstrating the continued need for adequate staffing for mHealth interventions to ensure quality implementation and monitoring. 2wT does not remove the need for VMMC client follow-up, but rather dramatically improves the efficiency of care. By no longer requiring in-person follow-up visits, 2wT reduces the time clinicians and staff at clinics need to spend on unnecessary clinic reviews. However, clinician oversight is still needed to monitor 2wT client healing and complete reviews for 2wT clients with desire or need for in-person care. In the SWD, during routine VMMC client volume, a dedicated 2wT-specific VMMC nurse spent an estimated 40% of their work time responding to clients with concerns via text message and phone call, if

**Table 4.** (A) Cost parameters and (B) estimated cost savings per client of 2wT vs. routine VMMC follow-up for extended implementation of 2wT.

A.

| Cost Inputs | Routine | 2wT | Source |
|---|---|---|---|
| General | | | |
| Number of clients | 4256 | 2586 | SWD |
| Average number of follow-up visits | 2.06 | 0.07 | |
| Number of moderate/severe AEs (%) | 0.13% | 0.60% | |
| Number of clients lost to follow-up | 342 | 93 | |
| Number of sites | 3 | 3 | |
| Duration of study (years) | 0.718 | 0.718 | |
| Annual duration of campaign period (weeks) | 4 | 4 | |
| Personnel | | | |
| Enrolled cadre level nurse wage (monthly) | $1,426.47 | $1,426.47 | Aurum Institute pay structure |
| IT/data manager wage (monthly) | – | $4,312.50 | I-TECH DIGI |
| Intern wage (monthly) | – | $284.96 | SWD |
| 2wT text messaging | | | |
| SMS system aggregated cost (monthly) | – | $132.66 | Africa is Talking, TextIT |
| Cost per SMS message | – | $0.01 | SWD |
| Average number of messages per client | – | 20 | SWD |
| Staff airtime cost (monthly) | – | $10.78 | |
| Post-op counseling | | | |
| Nurse review time (minutes) | 5 | 10 | RCT (assumption) |
| In-clinic follow-up visits | | | |
| Nurse review time in-clinic (minutes) | 5.45 | 5.45 | RCT (time-motion study) |
| Client tracing | | | |
| Number of phone call attempts | 3 | 3 | RCT |
| Length of tracing phone call (minutes) | 5 | 5 | |
| Number of home tracing attempts | 3 | 3 | |
| Mean distance to patient home (km) | 29.73 | 29.73 | |
| Liter of petrol (per km) | 0.06 | 0.06 | |
| Price of petrol (per liter) | $1.23 | $1.23 | Automobile Association of South Africa |

B.

| Annual number of VMMCs performed | Enrollment rate in 2wT | Cost savings: 2wT vs. routine |
|---|---|---|
| 10,000 | 40% | $0.06 |
| | **45%** | $0.00 – cost neutral |
| | 60% | -$0.29 |
| | 80% | -$0.46 |
| 100,000 | 40% | -$0.88 |
| | 60% | -$0.91 |
| | 80% | -$0.93 |

Abbreviations: I-TECH, International Training and Education Center for Health; DIGI, Digital Initiatives Group at I-TECH; RCT, randomized controlled trial; 2wT, two-way texting; SWE, stepped wedge expansion.

desired, directing them to return to their clinic for in-person follow-up. During high-enrolling campaign periods, a full time 2wT hub clinician was needed with support from an auxiliary intern at each 2wT enrolling site to complete critical activities like confirming 2wT enrollments on participant phones, alerting VMMC teams for tracing or referral tasks, and reconciling paper to digital records. 2wT makes VMMC follow-up more efficient and raises the

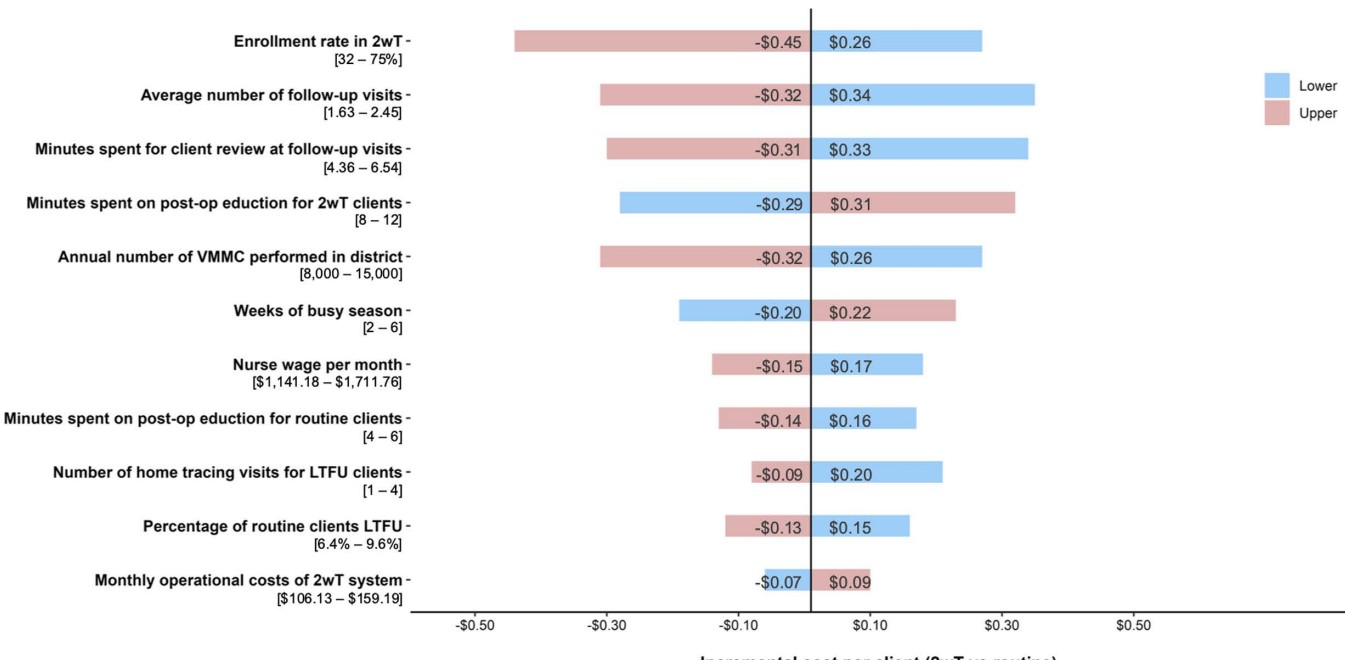

**Fig 2. One-way univariate sensitivity analysis of input factors comparing 2wT to routine care.** All variables that changed the incremental cost by more than $0.15 are shown. Colors correlate to input ranges of each parameter. (e.g., 32% enrollment rate in 2wT increases costs by $0.26/client while a 75% enrollment rate decreases costs by $0.45/client).

quality of post-operative care when implemented according to standard operating procedures. 2wT reduces, but does not remove, the need for clinical oversight and quality care assurances. During scale-up, personnel costs could be reduced by using a trained enrolled nurse, a lower category of nurse, instead of a professional nurse, as with additional training they can offer the same quality of follow up care.

By using the activity-based costing method, we provide clarity on the potential cost savings of 2wT while increasing the transparency around the costs of scaling up mHealth interventions. The lack of data on costs, especially in scale-up, remains a key barrier in implementing mHealth interventions, particularly in LMICs [9]. By separating the costs of implementation during the intensive phase of the SWD into routine-administration and research-specific costs, we demonstrate the variation of cost during the research-supported phase versus the cost to the funder while transitioning to independent management. Referring to recommendations on the scope and disaggregation of costs, we reported estimates using a micro-costing approach to increase accuracy and separated costs by phase to delineate the long-term costs of scale-up [9,20,21].

Our cost analysis of routine 2wT implementation also increases transparency on the true costs of mHealth interventions to better inform future decisions on the pace and scope of scale-up. Despite widespread interest and the rapid expansion of digital health technologies across SSA in the past decade, few interventions have been successfully transitioned beyond the pilot phase [22,23]. Among the barriers to successful mHealth expansion in LMICs are lack of consideration for infrastructure costs and lack of transparent information on the costs of mHealth implementation in routine settings [9,12]. First, we acknowledge that digital infrastructure costs necessary to support scale up were not included in this study. Quantification of infrastructure, including broader telecom or broadband costs, is complicated by difficulties applying standard program implementation cost estimation methods to a rapidly-evolving, and dynamically-priced, technology landscape [24,25].

However, the impact of infrastructure costs on cost per client estimates may not be as critical in South Africa where the reported rates of 3G coverage already reached 99.9% by 2021[26]. Second, in this analysis, we followed recommendations to increase transparency in the economic evaluation of digital health interventions by disaggregating costs across the development, implementation, and evaluation phases of 2wT operations in routine practice [9,20,25]. Our scenario analysis showed that integrating 2wT into routine clinical practice became cost-neutral at a typical VMMC program enrollment rate and demonstrated the possible cost savings with economies of scale as utilization increases. By detailing cost components and categories from real-world implementation, this analysis provides additional evidence to inform 2wT scale-up feasibility

Our study had several limitations. First, the cost savings of 2wT compared to routine in-person follow-up was estimated using data collected during the prior RCT, as data on post-operative visits and client tracing were not collected during the SWD. However, RCT tracing rates and time-motion data stemmed from both rural and urban clinics operated by the same implementing partner, reducing bias in tracing and provider time estimates. Second, the technology costs in 2wT changed over the study period; however, these changes are typical in the lifecycle of mHealth technology, including increasing monthly website-hosting or monthly data fees, indicating a challenge in costing mHealth interventions [21]. Finally, we likely underestimated the costs associated with client tracing for both 2wT and routine clients. Due to lack of data available from routine clinics surrounding on tracing efforts for clients with missed in-person follow-up visits, we only calculated tracing costs for clients identified as LTFU, assuming the maximum effort (3 phone calls and 3 home visits) had been used to attempt to locate them prior to being labeled as LTFU. We did not have data on clients who may have missed some visits but had been located using follow-up tracing and could not include these costs in the analysis. Therefore, our estimate may under value 2wT cost savings as routine clients were more likely to be LTFU and this analysis likely underestimates of the total cost spent on client tracing by routine clinics.

## Conclusion

Overall, evidence from this cost analysis of the 2wT approach in routine settings found that implementation would be cost-neutral to the payer with low enrollment and cost savings at scale. Combining with previous evidence that 2wT safeguards client safety while improving the quality of post-operative care, there are clear advantages to adoption of the opt-in 2wT approach for males who want to receive telehealth-based follow-up. Further scale-up would likely increase cost savings and dramatically reduce the burden of in-person reviews on clinics and clients.

## Supporting information

**S1 Table. Costing tool for 2wT implementation.** Modifiable Excel spreadsheet for 2wT costing during stepped wedge expansion study and for additional scenario analysis.
(XLSX)

**S2 Table. Consolidated Health Economics Evaluation Reporting Standards 2022 (CHEERS 2022) Checklist.**
(DOCX)

## Acknowledgements

The authors would like to thank the national and provincial Departments of Health of the Gauteng and North West; implementing partner, Right to Care, for enabling participation of the clinical teams; men who participated in the expansion study, and the Aurum Institute study implementation team for their dedication and skill in study implementation.

## Author contributions

**Conceptualization:** Molly Unsworth, Geoffrey Setswe, Jacqueline Pienaar, Caryl Feldacker.

**Data curation:** Molly Unsworth, Isabella Fabens, Tracy Dong, Caryl Feldacker.

**Formal analysis:** Molly Unsworth, Tracy Dong, Monisha Sharma.

**Funding acquisition:** Geoffrey Setswe, Jacqueline Pienaar, Caryl Feldacker.

**Investigation:** Khumbulani Moyo, Motshana Phohole, Nelson Igaba, Sizwe Hlongwane, Maria Sardini, Tracy Dong, Caryl Feldacker.

**Methodology:** Molly Unsworth, Isabella Fabens, Geoffrey Setswe, Jacqueline Pienaar, Calsile Makhele, Maria Sardini, Monisha Sharma, Caryl Feldacker.

**Project administration:** Geoffrey Setswe, Khumbulani Moyo, Jacqueline Pienaar, Calsile Makhele, Motshana Phohole, Maria Sardini, Felex Ndebele, Marrianne Holec.

**Resources:** Geoffrey Setswe, Khumbulani Moyo, Jacqueline Pienaar, Marrianne Holec.

**Software:** Molly Unsworth, Tracy Dong.

**Supervision:** Molly Unsworth, Geoffrey Setswe, Caryl Feldacker.

**Validation:** Molly Unsworth, Caryl Feldacker.

**Visualization:** Molly Unsworth, Isabella Fabens, Tracy Dong.

**Writing – original draft:** Molly Unsworth, Isabella Fabens, Caryl Feldacker.

**Writing – review & editing:** Molly Unsworth, Isabella Fabens, Geoffrey Setswe, Khumbulani Moyo, Jacqueline Pienaar, Calsile Makhele, Motshana Phohole, Nelson Igaba, Sizwe Hlongwane, Maria Sardini, Tracy Dong, Hannock Tweya, Felex Ndebele, Marrianne Holec, Caryl Feldacker.

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
