## [Decision Letter · Decision Letter 0]

27 Jan 2025

PGPH-D-24-02748

What does it cost to expand two-way texting for post-operative follow-up? A cost analysis in routine voluntary medical male circumcision settings in South Africa

Dear Dr. Feldacker,

Thank you for submitting your manuscript to PLOS Global Public Health. After careful consideration, we feel that it has merit but does not fully meet PLOS Global Public Health’s publication criteria as it currently stands. Therefore, we invite you to submit a revised version of the manuscript that addresses the points raised during the review process.

We look forward to receiving your revised manuscript.

Kind regards,

Damen Haile Mariam, MD, MPH, PhD

Academic Editor

Journal Requirements:

1. Please provide an Author Summary. This should appear in your manuscript between the Abstract (if applicable) and the Introduction, and should be 150–200 words long. The aim should be to make your findings accessible to a wide audience that includes both scientists and non-scientists. Sample summaries can be found on our website under Submission Guidelines:

https://journals.plos.org/globalpublichealth/s/submission-guidelines#loc-parts-of-a-submission.

Additional Editor Comments (if provided):

Reviewer 1:

Title -

- In the title, it is recommended that the study should reflect which health interventions were being compared. In this study it was the intervention (2wt) which boldly appeared - but not the comparator (routine care). This needs to be reviewed.

Background -

- It is a standard and requirement to explain whether the study is full economic evaluation or partial evaluation by developing Health Economics Analysis Plans (HEAP) for guiding the conduct of the economic evaluation as per the Consolidated Health Economic Evaluation Reporting Standards (CHEERS) checklist. Can the authors justify why this was not done?

Methods -

- Can the authors provide what type of sensitivity analysis was used? One-way? Two-way? This need to be specified and the rational for the choice.

Results -

- Describing the cost in ZAR is important as that of USD, and this has to be provided not only the exchange rate.

Other comments -

- The role of the funder in the identification, design, conduct, and reporting of the analysis was not clearly reported.

- There also needs for some editorial issues. The authors should note that "data" are plural.

Reviewer 2:

- Additional evidence that targets scaleup, if any, that would support your conclusion that “2wT should be expanded for eligible males across VMMC and other postoperative contexts in South Africa” would be essential.

- Does this conclusion serve clients/adults living in urban, rural, or both?

- Do you add a support of evidence that shows the current digital infrastructure of South Africa makes it easy for your conclusion to implement or scale up? At least you should provide evidence on the coverage of clients who can use a basic mobile phone with SMS capability.

Reviewers' comments:

Reviewer's Responses to Questions

**Comments to the Author**

1. Does this manuscript meet PLOS Global Public Health’s publication criteria ? Is the manuscript technically sound, and do the data support the conclusions? The manuscript must describe methodologically and ethically rigorous research with conclusions that are appropriately drawn based on the data presented.

Reviewer #1: Yes

Reviewer #2: Yes

2. Has the statistical analysis been performed appropriately and rigorously?

Reviewer #1: Yes

Reviewer #2: Yes

3. Have the authors made all data underlying the findings in their manuscript fully available (please refer to the Data Availability Statement at the start of the manuscript PDF file)?

Reviewer #1: Yes

Reviewer #2: Yes

4. Is the manuscript presented in an intelligible fashion and written in standard English?

Reviewer #1: Yes

Reviewer #2: Yes

5. Review Comments to the Author

Reviewer #1: This paper estimates the cost of implementation and expanding two-way texting for post-operative follow-up in routine voluntary medical male circumcision settings in South Africa

My thanks to the authors as this paper addresses an important and understudies topic. The paper is well written clearly stating the gap in the literature that it is intended to fill. The methodology is well explained and appropriate for their objectives. I have a few comments that I believe will improve the paper and make it publishable.

1. In the title it is recommended that the study should reflect which health interventions were being compared. In this study it was the intervention (2wt) which was boldly appeared but not the comparator (routine care). This needs to be reviewed.

2. The background of the study was illustrated well. Provided the context for the study, the study question, and its practical relevance for decision making in policy or practice.

3. It is a pretty standard and requirement whether the study is full economic evaluation or partial developing Health Economics Analysis Plans (HEAP) for guiding the conduct of economic evaluation as per the Consolidated Health Economic Evaluation Reporting Standards (CHEERS) checklist.

Can the authors justify why this was not done?

4. I agree with the statistical analysis and the supplementary file provided .Can the authors provide

What type of sensitivity analysis was used? One-way? Two-way? This need to be specified and the rational for the choice

5. In the result section describing the cost in ZAR is important as that of USD and this has to be provided not only the exchange rate.

6. The role of the funder in the identification, design, conduct, and reporting of the analysis was not clearly reported.

Reviewer #2: Dear Authors,

Happy that I reviewed this paper and got it sound with RCT/SWD.

The following questions on the abstract, introduction, and background have to be elaborated to make the publication complete.

Additional evidence that targets scaleup, if any, that would support your conclusion that “2wT should be expanded for eligible males across VMMC and other postoperative contexts in South Africa” would be essential.

Does this conclusion serve clients/adults living in urban, rural, or both? Do you add a support of evidence that shows the current digital infrastructure of South Africa makes it easy for your conclusion to implement or scale up? At least you should provide evidence on the coverage of clients who can use a basic mobile phone with SMS capability.

6. PLOS authors have the option to publish the peer review history of their article (what does this mean? ). If published, this will include your full peer review and any attached files.

**Do you want your identity to be public for this peer review?** For information about this choice, including consent withdrawal, please see our Privacy Policy .

Reviewer #1: No

Reviewer #2: No

---

## [Editor Report · Decision Letter 1]

20 Mar 2025

Expanding two-way texting for post-operative follow-up: a cost analysis of the implementation and scale-up in routine voluntary medical male circumcision settings in South Africa

PGPH-D-24-02748R1

Dear Feldacker,

We are pleased to inform you that your manuscript 'Expanding two-way texting for post-operative follow-up: a cost analysis of the implementation and scale-up in routine voluntary medical male circumcision settings in South Africa' has been provisionally accepted for publication in PLOS Global Public Health.

Best regards,

Damen Haile Mariam, MD, MPH, PhD

Academic Editor